# Cerebral Vascular Reactivity in Frail Older Adults with Vascular Cognitive Impairment

**DOI:** 10.3390/brainsci9090214

**Published:** 2019-08-24

**Authors:** Sara G. Aguilar-Navarro, Alberto José Mimenza-Alvarado, Isaac Corona-Sevilla, Gilberto A. Jiménez-Castillo, Teresa Juárez-Cedillo, José Alberto Ávila-Funes, Gustavo C. Román

**Affiliations:** 1Geriatrics & Neurology Fellowship, Instituto Nacional de Ciencias Médicas y Nutrición Salvador Zubirán, Mexico City 14080, Mexico; 2Department of Geriatric Medicine, Instituto Nacional de Ciencias Médicas y Nutrición Salvador Zubirán, Mexico City 14080, Mexico; 3Epidemiologic and Health Service Research Unit, Aging Area, Mexican Institute of Social Security, National Medical Center Century XXI, Mexico City 06720, Mexico; 4Centre de Recherche Inserm, U1219, F-33076 Bordeaux, France; 5Methodist Neurological Institute, Houston Methodist Hospital, Houston, Texas, and Weill Cornell Medical College, New York, NY 77030, USA

**Keywords:** cerebral vascular reactivity, frailty, mild vascular cognitive impairment, neuropsychological tests, older adults, diabetes, transcranial doppler ultrasound

## Abstract

**Background**: Frailty, a state of increased vulnerability, could play a role in the progression of vascular dementia. We aim to describe the changes in cerebrovascular reactivity of older adults with frailty and vascular-type mild cognitive impairment (MCIv). **Methods**: This was a cross-sectional study. A comprehensive geriatric assessment, neuropsychological evaluation, and transcranial Doppler ultrasound (TCD) was performed on 180 participants who were allocated into four groups: healthy (*n* = 74), frail (*n* = 40), MCIv (*n* = 35), and mixed (frail + MCIv) (*n* = 31). ANOVA and Kruskal–Wallis tests were used for the analysis of continuous variables with and without normal distribution. Multinomial logistic regression was constructed to identify associated covariates. **Results**: Subjects in the mixed group, compared to healthy group, were older (75.0 ± 5.9 vs 70.3 ± 5.9 years; *p* < 0.001), showed lower education (9.3 ± 6.4 vs 12.2 ± 4.0 years; *p* = 0.054), greater frequency of diabetes (42% vs 12%; *p* = 0.005), worse cognitive performance (z = −0.81 ± 0.94), and reduced left medial-cerebral artery cerebrovascular reactivity (0.43 ± 0.42 cm/s). The mixed group was associated with age (odds ratio (OR) 1.16, 95% Confidence Interval (CI) = 1.06–1.27; *p* < 0.001), diabetes (OR 6.28, 1.81–21.84; *p* = 0.004), and Geriatric Depression Scale (GDS) score (OR 1.34, 95% CI = 1.09–1.67; *p* = 0.007). **Conclusions**: Frailty among older adults was associated with worse cognitive performance, diabetes, and decreased cerebral blood flow.

## 1. Introduction

Frailty is a clinical state in which there is an increase in an individual’s vulnerability for developing increased dependency and/or mortality when exposed to a stressor. Frailty can occur as the result of a range of diseases and medical conditions.

Frailty is an important medical syndrome, in which there is an increase in an individual’s vulnerability to environmental stressors and an increased risk and accumulation of health-related problems, hospitalizations, need for long-term care, and death [1,2]. Despite numerous studies, a clear consensus on the definition of frailty is not stated [3]. Mild cognitive impairment (MCI) is the intermediate state between normal cognitive aging and dementia [4]. Up to 21% of MCI cases are due to vascular pathology [5]. Vascular-type mild cognitive impairment (MCIv) is a cognitive disorder caused by a spectrum of contributing vascular pathologies of multiple origins, not exclusively ischemic or hemorrhagic [6].

Previous studies have demonstrated a bidirectional relationship between frailty and MCI, where a frail subject is at greater risk of MCI and vice versa. Since both clinical situations are often considered reversible, a better understanding of the relationships between these two syndromes could allow for the establishment of interventions for their prevention and management [7]. In 2010, Gobbens et al. proposed the inclusion of cognitive impairment as part of the operational definition of frailty, demonstrating that up to 50% of frail subjects have some type of cognitive complaint [8]. 

The physiopathology that connects these two entities has been explained through several mechanisms. Aging induces cerebral vascular changes, such as decreased blood vessel lumen, loss of vasodilatation, reduced capillary density, collagen deposition, alterations of the blood–brain barrier (BBB), and endothelial dysfunction. These also occur in both frail older adults and patients with cardiovascular diseases. Cerebral small-vessel disease, including white matter hyperintensities and lacunar infarcts, are linked with cognitive decline and have been associated with vascular aging [9,10]. Recently, transcranial Doppler ultrasound (TCD) has emerged as a non-invasive, available, and relatively affordable technology that allows for the real-time evaluation of hemodynamic parameters. Studies have shown a consistent association with cognitive impairment [11]. TCD is a validated technology that can be used for the assessment of cerebral blood flow (CBF) velocities, reactivity, and cerebral autoregulation in order to determine how the changes in the neurovascular unit contribute to cognitive impairment and the aging process [12].

The aim of this study is to describe the changes in cerebral vascular reactivity among older adults with frailty and MCIv.

## 2. Materials and Methods

### 2.1. Participants

A cross-sectional study was conducted at an outpatient memory clinic of a tertiary level hospital in Mexico City. A total of 180 older adults aged 60 or older were recruited. Each subject underwent a clinical and neuropsychological assessment, TCD, and cerebral magnetic resonance imaging (MRI) between June 2017 and June 2018. On the basis of the results of the evaluations, participants were allocated into four mutually exclusive groups: (1) healthy, (2) frail, (3) MCIv, or (4) with mixed diagnosis (frail + MCIv). For the present study, subjects with major depressive syndrome, non-Alzheimer’s disease (AD) dementias, or other neurological disorders, including structural cerebral lesions that could affect cognitive function, acute stroke, brain tumors, normal pressure hydrocephalus, delirium, vitamin B_12_ or folic acid deficiency, partially treated hypothyroidism, severe heart failure, recent traumatic brain injury, alcoholism, acquired immune deficiency syndrome (AIDS), or cancer were excluded. All subjects gave their informed consent for the inclusion before they participated in the study. The study was conducted in accordance with the Declaration of Helsinki, and the protocol was approved by the local ethics committee.

### 2.2. Frailty Syndrome

Frailty was established according to the criteria proposed by Fried et al. [1] and validated in the Mexican population [13] as follows: The presence of (1) unintentional weight loss (≥5 kg in the previous year). (2) Poor endurance and energy (exhaustion) was established according to two questions from the Center for Epidemiological Studies Depression scale (CES-D): “I felt that everything I did was an effort” and “I could not get going”. Participants were asked “How often in the last week, did you felt this way?”, and those who answered “a moderate amount of the time” or “most of the time” to either of these questions were categorized as frail for this component [14]. (3) Weakness, defined by a reduced hand grip strength, measured with the use of a dynamometer, stratified by body mass index (BMI) and sex. (4) Slowness, established by a reduced gait speed in a 4 m distance, stratified by sex and height. (5) Reduced physical activity when participants performed in the lowest quintile of the Physical Activity Scale for the Elderly (PASE). Subjects were considered frail when they fulfilled three or more criteria and non-frail when one or two criteria were met [1].

### 2.3. Vascular Mild Cognitive Impairment (MCIv)

Functional and neuropsychological evaluations were performed in order to establish each subject’s cognitive status, including the following tests:Mini-Mental State Examination (MMSE) for the evaluation of global cognition (scores between 24–30 were included) [15].Clinical Dementia Rating Scale for the evaluation of cognitive and functional performance (score = 0.5 were included) [16].The Katz Index for activities of daily living (ADL) and the Lawton and Brody Index for the instrumental activities of daily living (IADL) were used to determine functional status [17,18]. If participants stated that they were unable to perform one or more activities without help, they were considered as having IADL or ADL disability.The brief neuropsychological evaluation in Spanish (NEUROPSI) developed to evaluate a wide range of cognitive domains including orientation, attention and concentration, language, memory, executive functions, lecture, writing and calculus was used. Z-scores were determined for each cognitive domain, and were adjusted for age and educational level [19].The diagnosis of non-amnestic MCI was established according to Petersen’s criteria, which included: (a) memory complaint or increased forgetfulness, (b) memory impairment (according to standardized neuropsychological tests), (c) ADL not affected by cognitive deficits, and (d) absence of dementia [4].

Additionally, to determine the vascular etiology of MCI, the definition of the Latin American Delphi Consensus on Vascular Cognitive Impairment was used. There are two critical areas for the diagnosis of vascular dementia: one, the certainty of the presence of a cognitive disorder, and two, the determination that vascular disease is the dominant, if not the only pathology, accounting for the cognitive disorder [20]. 

All participants underwent brain MRI using a standard protocol. Images were obtained with a 1.5-T Magnetic Resonance Scanner (Siemens^®^ Medical Systems, Mexico City, Mexico). Sequences included a whole-brain T2-weighted, T1-weighted and T2*-weighted gradient-recalled echo, fluid attenuation inversion recovery (FLAIR), and diffusion. A neuroradiologist, blinded to the subject’s clinical information, evaluated the presence of vascular burden using the Fazekas scale (FS) [21].

### 2.4. Normal Cognition 

Cognitively healthy patients were those without memory complaints and with normal neuropsychological tests, according to age and educational level [22].

### 2.5. Transcranial Doppler Ultrasound (TCD)

In order to determine CBF velocities, cerebrovascular reactivity (CVR) and the pulsatility index (PI), the middle cerebral artery (MCA) was bilaterally insonated through the trans-temporal window with the use of a 2 MHz transducer (DWL Doppler box, DTC digital, Compumedics^®^, Germany). All measurements were recorded. Finally, a breath-holding test (BHT) was used. Participants were requested to hold their breath for at least 30 s to reach a maximal flow velocity (MFV). CVR was calculated as a percentage of baseline MFV and absolute changes by subtracting the baseline values from the maximum MFV during the BHT task as follows: CVR = ((MFV_BHT_ − MFV_rest_)/MFV_rest_) × 100 [10]. A single operator performed all TCD assessments. A CVR score lower than 0.5% was considered abnormal, and it was managed as a continuous variable throughout the study. Participants that could not perform the BHT or did not have an adequate acoustic bone window were excluded.

### 2.6. Socio-Demographic and Clinical Variables

Information about age, sex, educational level (in years), body mass index (BMI (kg/m^2^)), smoking status, alcohol use, history of chronic diseases such as diabetes, hypertension, hypothyroidism, dyslipidemia, and use of visual of hearing aids, was also recorded. Depressive symptoms were evaluated with the Geriatric Depression Scale (GDS; 15-item version) [23].

### 2.7. Statistical Analyses

For the analysis of socio-demographic and clinical variables, means, standard deviations (SD), or their nonparametric equivalents (medians and interquartile ranges) were given. The one-way ANOVA test was used for continuous variables with normal distribution and the Kruskal–Wallis test was used for those without normal distribution. The chi-squared test was used for categorical data. To compare the different domains of cognitive function assessed by the neuropsychological battery, Z-scores were determined. The changes in vascular reactivity were analyzed by comparing clinical health parameters, cognitive performance, and frailty status. Finally, multinomial logistic regression models were constructed to identify covariates associated with the frail, MCIv, and mixed groups. Odds ratio (OR) was chosen as the measure to express the strength of the associations. A *p*-value of 0.05 or less was considered significant, and the confidence interval (CI) was 95%. The analysis was performed using STATA SE v.12.0 (Stata Corp. College Station, Texas, USA).

## 3. Results

Table 1 summarizes the socio-demographic characteristics of the study sample. Mean age was 73 ± 6.6 years, 75% were female, and the mean educational level was 11.5 ± 5.3 years. Hypertension (47%), dyslipidemia (33%), and diabetes (24%) were the most frequent chronic diseases. Smoking history was present in 48%. From a total of 180 participants, 41% (*n* = 74) were healthy, 22% (*n* = 40) frail, 19% (*n* = 35) MCIv, and 17% (*n* = 31) had a mixed diagnosis (frail + MCIv).

Participants in the mixed subgroup were older (*p* < 0.001), had a higher frequency of diabetes (*p* = 0.005), and had a higher score in the GDS 15-item depression scale (*p* < 0.001). When comparing MMSE evaluations, the mixed diagnosis group had a lower score (*p* = 0.016). Table 1 shows the cognitive performance scores. The mixed subgroup showed greater difference in the memory and executive function domains, as well in global scores (*p* ≤ 0.001). Differences were also found between the healthy, frail, and MCIv groups regarding execute function (*p* ≤ 0.001).

Of the 180 participants, 27 (15%) were excluded because of the absence of a trans-temporal window to obtain TCD registration or because of the patient’s inability to perform the BHT. Of the participants, 62 (40%) were healthy, 34 (22%) frail, 34 (22%) had MCIv, and 23 (15%) had a mixed diagnosis. The mean blood flow velocity (MVF) of the right middle cerebral artery (R-CMA) was 44.8 ± 11.2 cm/s. The mean CVR of the R-CMA was 0.50 ± 0.57 cm/s, and of left middle cerebral artery (L-CMA) was 44.4 ± 11.5. The mixed diagnosis had lower mean values in MVF, R-CMA, and L-CMA, with 40.7 ± 9.1 cm/s (*p* = 0.004), 0.40 ± 0.40 cm/s (*p* = 0.067), and 0.43 ± 0.42 cm/s (*p* = 0.051), respectively (Table 2).

Table 3 shows the correlates of the frail, MCIv, and mixed groups. The multinomial logistic regression model identified the age (OR = 1.16, 95% CI = 1.06 to 1.27; *p* < 0.001), prevalence of diabetes (OR = 6.28, 95% CI = 1.81 to 21.84; *p* = 0.004), the GDS score (OR = 1.34, 95% CI = 1.09 to 1.67; *p* = 0.007), and BMI (OR = 1.17, 95% CI = 1.00 to 1.36; *p* = 0.053) that were independently associated with the mixed group.

## 4. Discussion

In the present study, the presence of frailty and MCIv showed a decrease in both CVR values and mean CBF velocities. Arterial stiffness and endothelial dysfunction, together with reduced CBF velocities, are age-related changes that have been previously associated with cognitive decline [24]. Catchlove et al. compared CVR between young and healthy older adults and its relationship with cognitive performance. The study reported that CVR in temporal regions of the brain decreases with age, a situation that could impact cognitive function directly, especially in attention and memory domains [25]. Another study by Shim et al. reported a reduction in both mean CBF velocities and CVR, which have a negative impact on cognitive function [10]. The reduction of peripheral vascular function that occurs as a part of the senescence process is another important aspect to consider. Cispo et al. compared vascular changes in subjects below 45 and over 65 years of age, and found that endothelial dysfunction and arterial stiffness, measured through vascular reactivity, were present in the older age group [26]. A gradual decreasing effect of both parameters in the healthy, MCIv, and frail subgroups was observed in our study. Abnormalities observed in the CVR in patients with cognitive disorders could be explained by arteriolar stiffness caused by the deposition of cerebral amyloid in the leptomeningeal arterioles (also called cerebral amyloid angiopathy), hyaline arteriolosclerosis, and microvascular endothelial dysfunction [27].

The cellular and molecular mechanisms that contribute to vascular aging are complex. Endothelial dysfunction could be related to the process of oxidative stress, in which insulin growth factor 1 (IGF-1) deficiency and inflammation leads to a lower nitric oxide bioavailability [25]. Lipecz et al. showed that neurovascular coupling, measured by diameter changes in the retinal arterioles’ response to light, was decreased in older patients when compared to the younger population [28]. 

Avila-Funes et al. showed that 22% of frail patients had cognitive impairment. It can be argued that in those who complied with the frail phenotype, cognitive impairment added a predictive value for poor outcomes [29]. In 2010, Raji et al. conducted a 10 year follow up longitudinal study including more than 900 Mexican-American participants, with the aim of finding an association between cognitive status and the development of frailty. Those with an MMSE score <21 had a higher risk of developing frailty (OR = 1.09, CI 95% 1.01–1.17, *p* = 0.02) even after adjusting for comorbidities (OR = 1.09, CI 95% 1.00–1.19, *p* = 0.03) [30]. In our study, 17% of subjects had both comorbidities. Furthermore, we observed that subjects that shared both frailty and MCIv were older, had a lower MMSE score, and had a higher GDS-15 item version score. Participants had a lower performance in executive functions, and the mixed subgroup had even lower scores in the global cognitive evaluation, with greater involvement in memory, reading attention, and concentration domains. 

CVR is a parameter for further development in this population, as lower values could indicate cognitive decline in frail patients. As well as this, frailty and dementia were more associated to vascular-type dementia than to Alzheimer’s disease (5.2 vs 2.1 times, respectively) [31]. Whether CVR could help differentiate one from another is a potential future research development. 

Regarding cardiovascular risk factors, this study shows that the presence of type 2 diabetes mellitus was mostly associated with cognitive impairment. Results shown in the Rotterdam study support our results [32]. Similarly, Wagner et al., demonstrated that in a 14-year follow-up, only type 2 diabetes mellitus prevailed as a factor attributable to the development of dementia [33]. 

The main strength of our study was that it included elderly patients with mild vascular cognitive impairment and frailty, and that all patients had an evaluation of the microcirculation through TCD. To our knowledge, this is the first study in the Mexican population that describes cognitive domains in subjects with MCI and frailty. Among the weaknesses of this study, we must mention its cross-sectional nature, which makes it difficult to establish a causal relationship between the conditions studied. As well as this, selection bias may have also been possible, as patients from a tertiary hospital memory clinic have greater cognitive decline compared to the overall population. In addition, diagnosis of comorbidities was based only on clinical evaluations. Our results must be interpreted with caution and require confirmation with different study populations.

## 5. Conclusions

Our study demonstrated that patients with frailty and MCIv had alterations in microcirculation measured through cerebral vascular reactivity, suggesting that small vessel disease is present in early stages of cognitive decline. Frailty among older adults was associated with worse cognitive performance and decreased cerebral blood flow. Our study shows that in frail subjects, age-related cerebrovascular reactivity assessments might be convenient, in addition to cognitive evaluations and risk factor control. Longitudinal studies are necessary to determine the causal relationship between these two entities, in order to elucidate if the presence of frailty accelerates the development of dementia.

## Figures and Tables

**Table 1 brainsci-09-00214-t001:** Sociodemographic characteristics, clinical status, and cognitive performance of the study sample.

Variables	Total	Healthy	Frail	MCIv	Mixed (Frail + MCIv)	*p*-Value
	*n* = 180	*n* = 74	*n* = 40	*n* = 35	*n* = 31	
Age, mean (SD) ^C^	72.99 (±6.6)	70.3 (±5.9)	75.2 (±6.8)	74.2 (±6.8)	75.0 (±5.9)	<0.001
Female (%)	135 (75)	63 (85.1)	31 (77.5)	20 (57.1)	21 (67.7)	0.012
Educational level, mean (SD) ^B^	11.53 (±5.3)	12.2 (±4.0)	12.2 (±5.5)	10.9 (±6.1)	9.3 (±6.4)	0.054
MMSE, mean (SD) ^E^	28.2 (±1.8)	28.5 (±1.6)	28.8 (±1.1)	27.7 (±2.2)	27.4 (±2.3)	0.016
GDS (SD) ^C^	2.6 (±2.6)	1.7 (±1.8)	2.9 (±2.6)	2.9 (±2.7)	4.1 (±3.4)	<0.001
Hypertension (%)	84 (46.7)	30 (40.5)	20 (50)	17 (48.5)	17 (54.8)	0.536
Dyslipidemia (%)	59 (32.8)	21 (28.3)	12 (30)	13 (37.1)	13 (41.9)	0.518
Hypothyroidism (%)	24 (13.3)	5 (6.7)	6 (15)	9 (25.7)	4 (12.9)	0.057
Smoking Status (%)	85 (47.2)	38 (51.3)	17(42.5)	21 (52.5)	9 (22.5)	0.122
Alcoholism (%)	85 (47.2)	42 (56.7)	13 (32.5)	18 (51.4)	12 (38.7)	0.073
Diabetes (%) ^C^	44 (24.4)	9 (12.1)	10 (25)	12 (34.2)	13 (41.9)	0.005
Use of visual aid (%)	141 (78.3)	56 (75.6)	32 (80)	28 (80)	25 (80.6)	0.983
Use of hearing aid (%)	19 (10.6)	3 (4.0)	5 (12.5)	5 (14.2)	6 (19.3)	0.099
Katz–ADL (SD)	5.7 (±0.46)	5.8 (±0.34)	5.7 (±0.49)	5.7 (±0.42)	5.6 (±0.67)	0.455
Lawton and Brody–IADL (SD) ^B^	7.3 (±1.3)	7.6 (±1.1)	7.2 (±1.1)	6.9 (±1.4)	7.1 (±1.6)	0.013
BMI (SD)	25.3 (±3.6)	24.8 (±3.5)	24.6 (±3.3)	25.9 (±2.8)	26.7 (±4.6)	0.076
Cognitive performance						
Orientation	0 (0)	0 (0)	0 (0)	0 (0)	0 (0)	----
Attention and concentration ^B,C,D,E^	−0.01 (0.9)	0.49 (0.86)	0.14 (0.82)	−0.49 (−0.97)	−0.77 (0.78)	≤0.001
Memory ^B,C,D,E^	0.01 (0.09)	0.47 (0.59)	0.16 (0.90)	−0.49 (0.93)	−0.77 (1.27)	≤0.001
Language ^B,D^	−0.01 (0.99)	0.10 (0.51)	0.24 (0.1258)	−0.35 (0.48)	−0.13 (0.12)	0.055
Visuospatial/vasoconstrictive abilities	−0.01 (1)	0.19 (1.50)	−0.08 (0.20)	−0.21 (0.49)	−013 (0.25)	0.207
Writing/reading ^B,C^	−0.03 (1.07)	0.23 (0.53)	0.5 (0.75)	−0.34 (1.42)	−0.40 (1.65)	0.015
Executive function ^B,C,D^	0.001 (1)	0.39 (0.88)	0.18 (0.81)	−0.54 (1.18)	−0.51 (0.96)	≤0.001
Total score ^B,C,D,E^	0.001 (0.99)	0.51 (0.62)	0.24 (0.95)	−0.63 (0.95)	−0.81 (0.94)	≤0.001

MCIv = vascular mild cognitive impairment, MMSE = mini mental state examination, GDS = Geriatric Depression Scale, ADL = activities of daily living, IADL = instrumented activities of daily life, BMI = body mass index. ^A^ healthy vs frail; ^B^ healthy vs MCIv; ^C^ healthy vs mixed; ^D^ frail vs MCIv; ^E^ frail vs mixed. Data are shown in means and standard deviations (SD). Analysis was performed with a Kruskal–Wallis or chi-square test.

**Table 2 brainsci-09-00214-t002:** Transcranial Doppler ultrasound TCD parameters for the middle cerebral artery vascular reactivity in healthy, frail, MCIv and mixed subgroups.

TCD Variables	Total *n* = 153	Healthy *n* = 62	Frail *n* = 34	MCIv *n* = 34	Mixed (Frail + MCIv) *n* = 23	*p*-Value
R-MVF cm/s ^A, B^	44.8 (±11.2)	48.4 (±11.1)	44.1 (±10.5)	41.5 (±9.1)	40.7 (±9.1)	0.004
R-Post-Apnea MVF cm/s	49.3 (±14.9)	51.8 (±15.6)	46.1 (±15.7)	44.7 (±13.1)	43.5 (±13.9)	0.464
L-MVF ^A, B^	44.4 (±11.5)	47.3 (±12.1)	43.4 (±12.2)	40.7 (±9.7)	39.1 (±9.9)	0.061
L-Post-Apnea MVF cm/s	47.2 (±15.3)	50.1 (±16.2)	46.6 (±18.3)	43.2 (±11.6)	42.3 (±12.0)	0.173
R-CVR cm/s ^A, B, C^	0.50 (±0.57)	0.60 (±0.61)	0.50 (±0.71)	0.42 (±0.38)	0.40 (±0.40)	0.067
L-CVR cm/s ^A, B, C^	0.50 (±0.61)	0.58 (±0.63)	0.51 (±0.76)	0.44 (±0.51)	0.43 (±0.42)	0.051
R-PI ^A, B^	0.90 (±0.20)	0.84 (±0.16)	0.90 (±0.22)	0.94 (±0.22)	0.97 (±0.23)	0.068
L-PI ^A, B^	0.91 (±0.21)	0.89 (±0.18)	0.90 (±0.24)	0.94 (±0.24)	0.94 (±0.16)	0.077

R: right-middle cerebral artery, L: left-middle cerebral artery, CVR: cerebrovascular reactivity, PI: pulsatility index, MVF: mean blood flow velocity. Analysis was performed with a Kruskal–Wallis or chi-square test. ^A^ healthy vs frail; ^B^ healthy vs MCIv; ^C^ healthy vs mixed; ^D^ frail vs MCIv; ^E^ frail vs mixed. Values are presented means ± standard deviation (SD).

**Table 3 brainsci-09-00214-t003:** Association of frail, MCIv and mixed groups and their covariates using a multinomial logistic regression analysis.

Variables	OR	95% CI	*p-Value*
Frail (*n* = 36)			
Age (years)	1.13	1.05–1.21	<0.001
Diabetes	2.23	0.71–7.04	0.170
GDS score	1.26	1.03–1.53	0.024
BMI	0.98	0.86–1.12	0.767
MCIv (*n* = 34)			
Age (years)	1.12	1.04–1.21	0.002
Diabetes	3.56	1.16–10.93	0.027
GDS score	1.21	0.99–1.48	0.062
BMI	1.10	0.96–1.26	0.156
Mixed (Frail + MCIv) (*n* = 24)			
Age (years)	1.16	1.06–1.27	<0.001
Diabetes	6.28	1.81–21.84	0.004
GDS score	1.34	1.09–1.67	0.007
BMI	1.17	1.00–1.36	0.053

Reference group is represented by healthy participants. OR: odds ratio; CI: confidence intervals; MCIv: vascular mild cognitive impairment; BMI: body mass index; GDS: Geriatric Depression Scale.

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
