# Peer review of "Cerebral Vascular Reactivity in Frail Older Adults with Vascular Cognitive Impairment"

_brainsci, 2019, doi:10.3390/brainsci9090214_

Round 1

Reviewer 1 Report

Thank you for the opportunity to review the manuscript entitled "Cerebral vascular reactivity in frail older adults with vascular cognitive impairment".  This is a very interesting cross-sectional study which was aimed to investigate the changes in cerebral vascular reactivity of older adults with frailty and vascular-type mild cognitive impairment (MCIv). In univariate comparisons the authors found that compared to the healthy group, subjects in the mixed diagnosis subgroup were older, showed lower educational level, higher frequency of diabetes, worse global cognitive performance and reduced cerebral vascular reactivity in the left medial-cerebral artery.

Please follow the comments of the reviewer:

-          There are some unclear aspects in the methods.  No extensive information is provided regarding the sampling procedures. What was the response rate?  Please add the information.  A selection bias may be present. 

 -          The results may be difficult to generalize to other populations, since the patients were recruited from a tertiary level Hospital. Is there any information regarding length of stay, in-hospital mortality, cause of hospitalization.  All patients admitted to a hospital have some risk for temporary cognitive decline or frailty dependent the admitting diagnosis. You may consider to compare and the primary diagnoses and the in-hospital outcomes between frailty, MCI and health groups.

-          You may consider using a multinomial logistic model to identify covariates associated with frail, MCIv, and mixed  patients. The categories of the outcome variable may be frail, MCIv, and mixed versus health as reference category. I think it would be interesting, despite of the small sample size.

-          What is the number of participants with pre-frailty?  The authors may also want to show analysis regarding pre-frail patients?

-          Add to conclusions what are the clinical implications of the study?

Author Response

 Observations Response Cahnges in the manuscrpt

There are some unclear aspects in the methods.  No extensive information is provided regarding the sampling procedures. What was the response rate?  Please add the information.  A selection bias may be present. 

Thank you for the observation. Our memory clinic has an established protocol, where all the clinical information and cognitive domains are assessed equally. We included the first 180 patients from June 2017 to June 2018 that accepted the described protocol. (We evaluate around 900 patients per year).

From these 180, 153 did complete the protocol. We had 27 drop outs. Our response rate was of 85%. The sample was selected by convenience.

However, it is true that the selection may have bias. Patients seen in our clinic are referred for memory or cognitive problems and doesn’t reflect the overall population. We added these observations as weaknesses.

Changes in lines:

265-268

The results may be difficult to generalize to other populations, since the patients were recruited from a tertiary level Hospital. Is there any information regarding length of stay, in-hospital mortality, cause of hospitalization.  All patients admitted to a hospital have some risk for temporary cognitive decline or frailty dependent the admitting diagnosis. You may consider to compare and the primary diagnoses and the in-hospital outcomes between frailty, MCI and health groups.

For objectives of this study, a pre-frail group was not included. Prefrail patients were not analyzed separately.

If during the evaluation, patients were considered as pre-frail, they were excluded from the study and invitation to it.

Changes in line:

92-94

What is the number of participants with pre-frailty?  The authors may also want to show analysis regarding pre-frail patients?

Thank you.

We have added new conclusions.

Changes made at line:

273-275

You may consider using a multinomial logistic model to identify covariates associated with frail, MCIv, and mixed patients. The categories of the outcome variable may be frail, MCIv, and mixed versus health as reference category. I think it would be interesting, despite of the small sample size.

Thank you for the observation.

The model and results have been included.

Changes made at:

Statistical Analysis:

Lines 153-156

Results:

Lines 193-197

Reviewer 2 Report

In this study the authors aimed to describe the changes in cerebral vascular reactivity among older adults with frailty and vascular-type mild cognitive impairment (MCIv). The pathophysiological link between frailty and cognitive disfunction in the context of the aging cerebrovasculature has been gathering growing interest in the scientific community and is a topic of great clinical interest. This study was designed and conducted at the Memory Clinic of a tertiary level Hospital in Mexico City. The clinical population that was studied consisted of a total of 180 older adults aged 60 or older. Each subject underwent a clinical and neuropsychological assessment, transcranial Doppler, and cerebral magnetic resonance imaging (MRI). The groups were then separated in 1) healthy, 2) frail, 3) MCIv); or 4) mixed diagnosis 70 (frail+MCIv). For this study frailty was established according to the criteria proposed by Fried et al. and is described in the manuscript. The authors of the study find that participants in the mixed subgroup were older, had a higher frequency of diabetes, and had a higher score in the GDS-15 item depression scale. When comparing MMSE evaluation, mixed diagnosis had a lower score. The mixed subgroup showed greater difference in the memory, executive function and global score. Execute function showed differences between healthy, frail and MCIv group. The mixed subgroup had lower means in MVF, R-CMA and L-CMA with 40.7 ± 9.1 cm/s , 0.40 ± 0.40 cm/s and 0.43 ± 0.42 cms/s. The methods are well described, and the procedures carried out skillfully. This work could open the field to potential longitudinal clinical studies to determine the casual relationship between frailty and MCIv. The experimental design is carefully laid out and the methodological approach is clearly explained. This article findings provide supportive evidence of the growing and more appreciated understanding of the importance of the role of the cerebrovasculature in the context of aging, frailty and cognitive decline. The reviewer finds this article suitable for publication in Brain Sciences pending the revision of minor elements.

The reviewer offers the following critiques:

Major Concerns: none

Minor Concerns:

-          Can the authors discuss why no young individual were included in the study? Would it not be beneficial for comparison purposes to include a healthy young control?

-          Please check and clarify the whole funding section as it seems that it is erroneously pasted

-          The authors are invited to cite recent findings supporting the importance of studying the brain microvasculature in the context of aging and cognitive function (PMID: 31209739)

-          For the benefit of the Brain Sciences general readership, the reviewer the reviewer strongly encourages the authors to cite in their work recently published findings regarding potential cerebromicrovascular mechanisms underlying the age-related loss of cognitive function:

(PMID: 31015147, PMID: 31144244, and 31030329)

Author Response

Observation Response  Changes in the manuscript

Can the authors discuss why no young individual were included in the study? Would it not be beneficial for comparison purposes to include a healthy young control?

The study of frailty in Geriatrics has proven to be an age-related condition, to the decrease in homeostatic reserves due to aging itself, which is why we consider that to identify possible mechanisms linked to frailty and cognitive impairment, so include Young population would not respond to our research objective at this time.

Please check and clarify the whole funding section as it seems that it is erroneously pasted.

Thanks a lot.

We added the observation in line 290-291

The authors are invited to cite recent findings supporting the importance of studying the brain microvasculature in the context of aging and cognitive function (PMID: 31209739)

Thank you for the recommendations. We have added the most recent findings supporting cerebrovascular reactivity and aging.

Cispo, T.; Lipecz, A.; Fulop, G.; et al. Age-related

decline in peripheral vascular health predicts

cognitive impairment. Geroscience 2019,41,125-136.

Available online: doi:10.1007/s11357-019-00063-5

Lipecz, A.; Csipo, T.; Tarantini, S.; et al. Age-related impairment of neurovascular coupling responses: a dynamic vessel analysis (DVA)-based approach to measure decreased flicker light stimulus-induced retinal arteriolar dilatation in healthy older adults. Geroscience 2019: 1,9. Available online: doi:10.1007/s11357-019-00078-y

Changes made on lines:

227-231

And

236-240

For the benefit of the Brain Sciences general readership, the reviewer the reviewer strongly encourages the authors to cite in their work recently published findings regarding potential cerebromicrovascular mechanisms underlying the age-related loss of cognitive function

Thank you. Changes have been made.

The authors appreciate your observations. We have done a general revision of the manuscript. Also, we have added the name of “Dr. Teresa Juarez Cedillo” who contributed to the revision and statistical analysis.

Round 2

Reviewer 1 Report

Reviewer believes that the manuscript has improved. However, some attention to fine detail and sentence structure is warranted. Below are some suggestions.

You wrote that 180 patients were enrolled to the study and 27 did not complete the protocol. 153 patients completed TCD exam.  Please add the distribution of factors why these 27 patients did not enrolled. For instance the absence of an acoustic window, refused and etc.  Can you add one statement about the distribution of vascular pathologies which were found in MRI evaluation?

Author Response

Thanks for your comments, it was specified because 27 participants were excluded and added to the results.Thanks for your comments, it was specified because 27 participants were excluded and added to the results.

With regard to magnetic resonance imaging, we have decided to add information about this study only to establish the diagnosis of MCIv, for which the Fazekas scale was used to measure the degree of small vessel load. These comments and references were added.

Finally, we corrected editing details.

Thanks again!